# miRNA-Based Therapeutics in the Era of Immune-Checkpoint Inhibitors

**DOI:** 10.3390/ph14020089

**Published:** 2021-01-26

**Authors:** Florian Huemer, Michael Leisch, Roland Geisberger, Nadja Zaborsky, Richard Greil

**Affiliations:** 1Department of Internal Medicine III with Haematology, Medical Oncology, Haemostaseology, Infectiology and Rheumatology, Oncologic Center, Paracelsus Medical University, 5020 Salzburg, Austria; f.huemer@salk.at (F.H.); m.leisch@salk.at (M.L.); r.geisberger@salk.at (R.G.); n.zaborsky@salk.at (N.Z.); 2Salzburg Cancer Research Institute–Laboratory for Immunological and Molecular Cancer Research (SCRI-LIMCR), 5020 Salzburg, Austria; 3Cancer Cluster Salzburg, 5020 Salzburg, Austria

**Keywords:** miRNA, microRNA, immune-checkpoint inhibitor, immune-checkpoint blockade, predictive, ipilimumab, nivolumab, anti-PD-1, anti-PD-L1, anti-CTLA-4

## Abstract

MicroRNAs (miRNAs) are small non-coding RNAs that regulate gene expression by binding to complementary target regions on gene transcripts. Thus, miRNAs fine-tune gene expression profiles in a cell-type-specific manner and thereby regulate important cellular functions, such as cell growth, proliferation and cell death. MiRNAs are frequently dysregulated in cancer cells by several mechanisms, which significantly affect the course of the disease. In this review, we summarize the current knowledge on how dysregulated miRNAs contribute to cancer and how miRNAs can be exploited as predictive factors and therapeutic targets, particularly in regard to immune-checkpoint inhibitor therapies.

## 1. Introduction

MicroRNAs (miRNAs) are a family of small (~19–24 nucleotides in length), non-coding RNAs that regulate gene expression on a posttranscriptional level. This is achieved by binding to complementary regions on messenger RNA (mRNA) transcripts, which inhibits translation or initiates decay of the target transcripts, thereby limiting the number of expressed proteins. By this mechanism, miRNAs substantially regulate cellular protein profiles, affecting many important biological pathways such as differentiation, proliferation and apoptosis. miRNAs were first described in 1993 in the nematode *Caenorhabditis elegans*, which relies on downregulation of the protein LIN-14 for larval stage progression. This downregulation was dependent on the transcription of the gene lin-4, which was not translated into protein but was processed to small RNAs, which had multiple complementary binding sites on the 3′ untranslated region of the LIN-14 transcript. These observations led to the first concept of small RNAs (miRNAs) causing translational repression of mRNAs [1,2]. Subsequent studies resulted in the discovery of further miRNAs in *C. elegans* and also in other invertebrates and vertebrates, which suggested that miRNA-mediated posttranscriptional regulation of gene expression is likely more general than previously anticipated and conserved across many different species [3,4,5]. Currently, there is a total of 38,589 known hairpin precursor miRNAs from 271 organisms, which are listed in an online repository for miRNA sequences and annotation (http://mirbase.org/), with the human genome comprising 1917 annotated hairpin precursors with 2654 mature miRNA sequences [6]. However, the biological impact of most of these miRNAs is still elusive.

Generally, miRNA containing transcripts are transcribed by RNA polymerase II as primary (pri)-miRNA with several hundreds of nucleotides in length with 5′ cap structures and 3′ polyadenylation. The pri-miRNA is then processed into ~70- to 120-nucleotide-long hairpin precursor RNA (pre-miRNA) by a multiprotein complex called Microprocessor comprising the RNase type III enzyme Drosha and the RNA binding protein DiGeorge syndrome critical region gene 8 (DGCR8). The pre-miRNA is then exported to the cytoplasm by exportin 5 and processed into mature ~18- to 23-nucleotide-long RNA duplexes by the RNase III enzyme Dicer-1. The two resulting miRNA strands comprise a guide and a passenger strand, with the passenger being degraded and the guide strand associating with a multiprotein RNA-induced silencing complex (RISC) consisting of the transactivation response RNA binding protein (TRBP), Argonaute 2 (Ago2), protein kinase RNA activator (PACT), trinucleotide repeat-containing gene 6A (TNRC6A), and other RNA binding proteins. Finally, RISC is recruiting the guide strand to the 3′ UTR of the target mRNA to either induce degradation of the mRNA (in case of perfect base complementarity) or inhibit translation (in case of imperfect miRNA/mRNA base pairing). In either case, the net result is a decrease in protein abundance [7]. Apart from this well studied role of miRNAs in mRNA binding, also non-canonical functions of miRNAs have been documented. Most strikingly, recent studies provide evidence that miRNAs directly affect gene expression in the nucleus upon complementary binding to regulatory elements on genomic DNA. This interaction can regulate RNA polymerase activity as well as the methylation status of the gene promoter, eventually leading to increased or repressed gene transcription [8]. As for many miRNAs, important roles in developmental timing, patterning, cell differentiation, cell death, angiogenesis, cell proliferation and hematopoiesis have been identified, and a contribution of miRNA dysregulation to degenerative diseases such as cancer has been postulated. In the field of hematology and medical oncology, huge efforts have been made in recent years in order to identify plasma or tumor tissue-based miRNAs as predictive biomarkers, which provide information about the likelihood of response to anticancer therapies such as chemotherapy [9,10], anti-angiogenic therapy [11,12] as well as immunotherapy [13,14,15] as well as to identify miRNA candidates for therapeutic purposes.

## 2. miRNA Dysregulation in Cancer

The first evidence that miRNA dysregulation is involved in cancer came from studies in chronic lymphocytic leukemia (CLL). In CLL, a region within chromosome 13q14 is frequently deleted, whereupon it was suggested that it would contain a putative tumor-suppressor gene. Analysis of the minimal deleted region revealed that two miRNA genes, miR-15a and miR-16-1, are located at this genomic position [16,17]. Subsequent studies confirmed a tumor-suppressive role of these miRNAs, mostly by repressing the expression of BCL-2, an antiapoptotic protein. Furthermore, targeted deletion of miR-15b/16-2 in mice promotes the formation of B cell malignancies, supporting a direct role of these miRNAs in cancer [18]. Subsequently, mapping of miRNA genes to chromosomal regions revealed that miRNAs were frequently located at fragile sites and cancer-associated chromosomal aberrations, such as deletions, amplifications and rearrangements [19]. Consistently, the presence of such chromosomal aberrations was associated with dysregulated expression of the respective miRNA. Moreover, large-scale miRnome analyses by microarray technologies, examining expression patterns of the complete set of miRNAs, confirmed the aberrant expression of many miRNAs and the presence of specific miRNA signatures in many solid cancer entities [20]. In this regard, dysregulated miRNA expression in cancer is not solely due to genetic aberrations at chromosomal loci encoding miRNA genes but is also caused by the cancer-specific expression of transcription factors and epigenetic differences, as their expression is sensitive to miRNA-promoter methylation and histone modifications [21,22]. Moreover, altered miRNA expression levels can be attributed to the altered activity of factors important for miRNA biogenesis, such as Drosha, Dicer, DGRC8, Argonaute or Exportin 5. Indeed, increasing evidence shows the existence of many cancer-specific mutations within these genes and cancer-specific alterations in their expression levels, which is frequently prognostically relevant [23].

Aside from epigenetic differences, also epitranscriptomic differences were discerned between cancer and normal tissue. Particularly, modification of single bases within miRNAs by a process called RNA editing can affect absolute levels and target specificities of miRNAs [24]. Generally, RNA editing is a highly conserved posttranscriptional mechanism in metazoans, which comprises specific deamination of adenosine (A) to inosine (I) by ADARs (adenosine deaminases that act on RNA) and deamination of cytosine (C) to uracil (U) by apolipoprotein B mRNA editing catalytic polypeptide-like (APOBEC) enzymes. As I is a guanosine analog, A to I editing has the same effect as an A to G conversion, and hence, both types of RNA editing can change the protein sequence of genes, the stability of RNAs and also the target sequence of miRNAs [25]. RNA editing is an important epitranscriptome diversifier in normal tissue [26]; however, RNA editing is frequently dysregulated in cancer and associates with patient survival [27,28,29]. As ADARs prefer double-stranded RNA as a target, miRNAs are an ideal ADAR substrate. The binding of ADAR to miRNA cannot only change individual bases within the mature miRNA but can also affect processing efficiency. This is corroborated by findings in *Caenorhabditis elegans*, which showed that approximately half of the miRNAs had altered expression levels in ADAR mutant strains [30]. In cancer, altered miRNA editing does also alter the mRNA target specificity as shown in melanoma and CLL, thereby significantly affecting protein expression profiles [29,31].

## 3. Response Prediction to Immune-Checkpoint Inhibitors by miRNAs

The therapeutic concept of unleashing a pre-existing antitumor immune response has led to meaningful improvements in clinical outcomes across various tumor entities. In recent years, multiple negative regulatory pathways, so-called immune-checkpoints, such as, e.g., cytotoxic T-lymphocyte protein 4 (CTLA-4), programmed cell death protein 1 (PD-1), programmed death ligand 1 (PD-L1), T-cell immunoglobulin-3 (TIM-3), lymphocyte activation gene-3 (LAG-3) and T-cell immunoglobulin and ITIM domain (TIGIT) have been investigated and therapeutically targeted by monoclonal antibodies termed “immune-checkpoint inhibitors” (ICI) [32]. The therapeutic success of immune-checkpoint blockade has, in turn, led to the approval of a variety of ICIs as monotherapy or as combination therapy for a plethora of tumor entities by the Food and Drug Administration (FDA) [33]. However, only a subgroup of patients derives long-term clinical benefit from immune-checkpoint inhibition in biomarker-unselected tumors [34,35,36]. Although certain tumor tissue-based biomarkers such as PD-L1 expression as well as tumor mutational burden (TMB) are associated with an increased likelihood to respond to the immune-checkpoint blockade, PD-L1 scoring systems and defined TMB cutoff values are not uniformly applicable across tumor entities, and PD-L1 negativity or a low TMB do not exclude responses. In this regard, easily available pretreatment serum miRNAs could serve as a non-invasive diagnostic approach in order to select patients most likely deriving benefit from the immune-checkpoint blockade and sparing probable nonresponders adverse events. Table 1 summarizes clinical data on the predictive role of miRNAs in cancer patients undergoing therapeutic immune-checkpoint inhibition.

### 3.1. Non-Small Cell Lung Cancer (NSCLC)

The therapeutic success of ICIs in non-small cell lung cancer (NSCLC) has initially been demonstrated by an OS advantage of the anti-PD-1 monoclonal antibody (mAb) nivolumab over docetaxel in palliative second-line among squamous [37] as well as non-squamous [38] NSCLC patients. In the meanwhile, ICI monotherapy [39,40], combined with chemotherapy [41,42,43], as well as immune-checkpoint inhibitor combination therapy [44], have been established as front-line treatment strategies in advanced NSCLC without driver mutations. In search of predictive biomarkers, several groups investigated pre-ICI-treatment miRNAs and miRNA dynamics as predictors of clinical outcome.

Halvorsen et al. profiled serum miRNAs by next-generation-sequencing (NGS) prior to treatment initiation with nivolumab in 20 chemotherapy pretreated NSCLC patients (discovery set) and classified patients according to overall survival (OS) from ICI start (11 “good responders” with an OS > 6 months versus 9 “poor responders” with an OS < 6 months). Among 309 identified miRNAs, expression levels of seven miRNAs (miR- 215-5p, miR-411-3p, miR-493-5p, miR-494-3p, miR-495-3p, miR-548j-5p and miR-93-3p) statistically significantly differed between the two groups. Based on a score integrating the latter miRNAs, good responders to nivolumab could be differentiated from poor responders with a sensitivity of 100% and specificity of 77% and results were confirmed in a validation set of 31 pretreated NSCLC patients (sensitivity 71%, specificity 90%). The seven miRNA-based score proved to be independently associated with OS in multivariate analysis in the discovery set and validation set, respectively, whereas the miRNAs did not predict OS among NSCLC patients without immune-checkpoint blockade [13]. The latter findings argue for a predictive value and against a barely prognostic value of the identified miRNAs. The physiologic function of, e.g., miR-215 as a tumor-suppressor by inducing cell cycle arrest [45] and of miR-548j by regulating interferon (IFN)-mediated pathways [46] may explain the poor clinical outcome with nivolumab in patients with low expression levels.

Shukuya et al. analyzed miRNA profiles by NGS in plasma and plasma extracellular vesicles (EV) from advanced NSCLC patients prior to initiation of anti-PD-1 or anti-PD-L1 targeting therapy. Patients achieving a partial response or stable disease for at least six months were classified as responders, whereas patients with progressive disease during this time frame were defined as nonresponders. Statistically significant differences of 32 plasma miRNA levels (*p* = 0.0030–0.0495) and seven EV-associated miRNA levels (*p* = 0.041–0.0457) were found between 14 responders and 15 nonresponders. The authors could corroborate the predictive value of some of the miRNAs in a validation set of 21 NSCLC patients undergoing immune checkpoint blockade [47].

Peng et al. investigated plasma EV miRNA expression levels, and dynamics among epidermal growth factor receptor (EGFR)/anaplastic lymphoma kinase (ALK) wild-type (WT) advanced NSCLC patients during immune-checkpoint blockade with anti-PD-1 or anti-PD-L1 monotherapy. Among 25 patients evaluable for radiologic response assessment, five patients with a partial response and four patients with progressive disease at re-staging were identified and included in the final analysis (16 patients with stable disease were excluded). Patients had received zero to three prior systemic therapy lines. Baseline levels of miR-320d, miR-320c, miR-320b were significantly lower in responders to ICI therapy compared to nonresponders [48]. Previous reports on the impact of the miR-320 family in lung carcinogenesis and progression are conflicting. While Lei et al. reported an inhibitory effect of miR-320 on cell proliferation, migration and invasion in NSCLC [49], Fortunato et al. found that miR-320a secreted by non-epithelial cells promoted an M2-like protumorigenic phenotype in macrophages associated with lung cancer risk [50]. When evaluating EV miRNA level dynamics among responders, miR-125-5p levels were found to be considerably downregulated over time. Furthermore, a trend towards higher miR-125-5p levels prior to initiation of the immune-checkpoint blockade was reported among nonresponders compared to responders [48]. The latter findings may be explained by the role of miR-125-5p as a downregulator of γδT cell activation [51]. As stated by the authors [48], unfortunately, sequential blood samples for the evaluation of miRNA level dynamics among nonresponders were not available and correlations between miRNAs and established predictive factors such as PD-L1 status were not carried out.

Boeri et al. prospectively applied the RT–qPCR-based miRNA signature classifier (MSC) to 140 advanced NSCLC patients prior to the initiation of ICI therapy [14]. The MSC, which incorporates 24 miRNAs from whole blood samples, was previously confirmed to prognosticate OS among NSCLC patients that had been diagnosed within a low-dose CT lung cancer screening [52]. The majority of included patients received ICI monotherapy in first- or second-line. Due to technical issues, only 79% (*n* = 111) of included patients were classifiable according to the MSC risk level. Among evaluable patients, 23%, 46%, and 31% of patients were classified as MSC high, intermediate and low-risk, respectively. An overall response rate of 28% was observed among MSC intermediate/low-risk patients, whereas none of the MSC high-risk patients responded to ICI therapy (*p* = 0.0009). The association between the MSC risk classification based on whole plasma as well as the PD-L1 expression status on tumor cells based on tissue samples and clinical outcome (PFS and OS) was subsequently evaluated among patients with both parameters available. The MSC risk level (intermediate/low versus high, H: 0.35, *p* = 0.0026), PD-L1 expression on tumor cells (≥50% versus <50%, H: 0.35, *p* = 0.0006) and combined MSC and PD-L1 score (1–2 favorable markers versus 0 favorable markers, H: 0.25, *p* = 0.0006) remained independently associated with PFS in multivariate analysis. Similar results were reported for OS (MSC: intermediate/low versus high, H: 0.28, *p* = 0.0007; PD-L1 expression on tumor cells: ≥50% versus <50%, H: 0.43, *p* = 0.0211; combined MSC and PD-L1 score: 1–2 favorable markers versus 0 favorable markers, H: 0.28, *p* = 0.0034). In a subset of patients, longitudinal MSC risk levels followed response to ICI therapy. However, due to the limited number of patients in this subanalysis, conclusions must be drawn with caution [14].

Genova et al. isolated and profiled EV miRNAs from whole plasma samples of 174 pretreated, advanced NSCLC patients prior to nivolumab initiation. Two miRNAs, miR-208a-5p and miR-574-5p, which were overexpressed in patients with an OS of less than nine months (*p* = 0.0009, respectively), were identified, and the poor clinical outcome was confirmed in each of ten validation sets (10 random validation sets derived from the validation cohort of 49 patients). Patients displaying miR-208a-5p and miR-574-5p expression levels below the median achieved a substantially longer OS after initiation of immune-checkpoint blockade compared to those with expression levels above the median (*p* < 0.0001). Of note, in silico prediction identified genes involved in the TGF-beta and antigen-presenting pathway as targets of miR-208a-5p and miR-574-5p, thereby corroborating a causative role of both miRNAs in resistance to ICIs [53].

### 3.2. Gastric and Esophageal Cancer

The therapeutic immune-checkpoint blockade has been initially investigated among pretreated advanced gastric/gastroesophageal junction (GEJ) cancer patients, which has led to the approval of pembrolizumab in PD-L1 combined positive score (CPS) ≥1 patients for third-line or subsequent line [54] and in case of microsatellite instability-high (MSI-H) or mismatch-repair deficient (MMRD) tumors as second-line or subsequent line therapy [55] by the FDA. Nivolumab (regardless of PD-L1 expression status) [56] and pembrolizumab (PD-L1 CPS ≥ 10) [57] improved OS when prospectively compared to the investigator’s choice in pretreated advanced esophageal squamous cell carcinoma. Recently, encouraging data from the Checkmate-649 study [58] as well as the Keynote-590 study [59] demonstrated an OS benefit with the addition of an ICI to first-line chemotherapy in biomarker selected (according to PD-L1 CPS) advanced gastric/GEJ cancer and advanced esophageal cancer, respectively. Meanwhile, therapy response adapted adjuvant application of nivolumab among esophageal/GEJ cancer patients with residual pathologic disease after neoadjuvant chemoradiotherapy clinically meaningfully increased DFS, thereby potentially establishing a new standard of care [60]. In gastroesophageal cancer, only a few reports evaluating the role of miRNAs for response prediction to ICI, mainly in pretreated advanced disease, have been published so far.

Sudo et al. investigated the predictive value of 2565 miRNAs among 19 pretreated advanced esophageal squamous-cell carcinoma patients included in a single-arm phase 2 trial [61] before and during treatment with nivolumab [62]. Pretreatment serum samples were available among all included patients, and miRNA expression levels were analyzed using a 3D-gene human miRNA oligo chip. Low-baseline expression levels of miR-1233–5p were found among four out of five responders to nivolumab (AUC 0.84, 95% CI: 0.56–1.00). Among 19 patients, serum samples were also available in 17 patients after nivolumab application with a median time interval of 29 days (range: 27–56 days) between nivolumab administration and blood sample collection. Low expression levels of miR-6885-5p, miR-4698, and miR-128-2-5p after nivolumab initiation helped discriminating responders from nonresponders: 71% responders (AUC 0.93), 83% responders (AUC 0.97) and 83% responders (AUC 0.93), respectively. Unfortunately, to date, the physiologic role of the abovementioned miRNAs has not been reported so far. In a similar approach, Miyamoto et al. applied the abovementioned microarray to 20 pretreated advanced gastric cancer patients that had received nivolumab within the ATTRACTION-02 trial [63]. One miRNA prior to nivolumab application and one miRNA four weeks after the first nivolumab application were identified, which predicted response to anti-PD-1 therapy (AUC 0.88 and AUC 0.85), respectively. Furthermore, high expression levels of both miRNAs were associated with superior PFS, respectively (14.3 months versus 1.6 months, H: 0.19, *p* = 0.01; 5.6 months versus 1.6 months, H: 0.21, *p* = 0.01) [64]. However, the abovementioned miRNAs were not further specified in the preliminary report.

### 3.3. Melanoma

The anti-CTLA-4 targeting mAb ipilimumab was the first checkpoint inhibitor demonstrating a survival benefit in advanced pretreated unresectable melanoma [65], ushering the era of immuno-oncology. In phase 3 clinical trials, the anti-PD-1 antibodies nivolumab [66] and pembrolizumab [67], respectively, demonstrated an improved OS compared to ipilimumab in advanced melanoma patients regardless of the BRAF mutation status and as a consequence constitute a front-line standard of care. ICI combination therapy (ipilimumab + nivolumab) [66], as well as combined immune-checkpoint blockade with BRAF and MEK-inhibition in BRAF mutant melanoma (atezolizumab + vemurafenib + cobimetinib) [68], represent therapeutic options in selected cases according to the current National Comprehensive Cancer Network (NCCN) treatment guidelines [69]; however, the latter combinations have not yet demonstrated an OS benefit. Meanwhile, nivolumab [70], as well as pembrolizumab [71], have been FDA approved for adjuvant melanoma therapy in stage IIIB-IV and stage III, respectively. Previous reports have already demonstrated the potential of plasma-based miRNA panels to distinguish melanoma patients from individual healthy plasma donors [72,73].

Huber et al. measured the expression levels of myeloid-derived suppressor cell (MDSC)-miRNAs among 20 advanced melanoma patients (unresectable stage IIIC and IV) and 20 sex-matched healthy donors and found higher levels of all MDSC-miRNAs among melanoma patients. These miRNAs were shown to drive the conversion from monocytes into MDSC by melanoma EV as a putative mechanism of resistance to ICIs. The pretreatment baseline plasma expression levels of let-7e, miR-125a, miR-99b, miR-146b and miR-125b turned out to be associated with clinical outcome in univariate analysis. Based on the expression levels of the latter miRNAs, the authors separated advanced melanoma patients (*n* = 49) into low scores (0–1, 0 or only 1 increased miRNA, *n* = 28) and high scores (2–5, 2 to 5 increased miRNAs, *n* = 21). Patients with low scores undergoing immune-checkpoint blockade with either ipilimumab or nivolumab displayed a statistically significantly better OS compared to high score patients (*p* = 0.0031). However, the score could not separate melanoma patients with differing OS risk during tyrosine-kinase inhibitor (TKI) therapy (*p* = 0.7531), corroborating its predictive value exclusively for patients treated with ICIs [15].

Galore-Haskel et al. found that miR-222 suppresses intercellular adhesion molecule 1 (ICAM1) expression in human metastatic melanoma cancer cell lines and, in turn, cytotoxic T-lymphocyte mediated death. Expression levels of miR-222 in tumor tissue samples prior to ipilimumab initiation were statistically significantly higher in melanoma patients progressing on ICI therapy (*n* = 8) compared to those five patients deriving a clinical benefit from immune-checkpoint blockade (*p* = 0.001). The authors reported a trend towards a higher number of patients with high ICAM1 expression based on tumor tissues in the group benefitting from ipilimumab, however, without any differences in lymphocyte infiltration or spatial lymphocyte distribution [74].

Taken together, the abovementioned reports on the predictive role of mainly plasma-based pretreatment miRNAs or miRNA-panels in cancer patients undergoing immune-checkpoint blockade are encouraging but accompanied by several shortcomings. First, the sample size of the discovery sets in the respective studies was relatively small, and the majority of studies lack a validation set. Second, the physiologic as well as the pathophysiologic role of several identified putative predictive miRNAs has not been clarified yet. As a consequence, discriminating a predictive value from a prognostic value may be challenging but could be overcome by investigating the impact of miRNAs in control groups not receiving ICIs [13,15]. Third, systemic pretreatment, as well as the time interval between prior systemic therapy and start of ICI therapy, might hamper interpretation of miRNA expression levels before ICI start. Fourth, miRNA expression (levels) depend on the tumor entity; therefore, findings are not applicable across various tumor types. Fifth, the heterogeneity of applied methodologies in the abovementioned studies highlight the challenge to standardize diagnostic approaches. Furthermore, the retrospective character of these biomarker studies is a major drawback; therefore, conclusions must be drawn with caution.

## 4. miRNAs as Therapeutic Adjuvant for Immune-Checkpoint Inhibitors

As outlined in the section above, miRNAs can aid in predicting response to ICI treatment and play a relevant role in resistance to immunotherapy. Therefore, it is very tempting to explore miRNAs as therapeutic agents in order to augment responses to ICIs.

Generally speaking, miRNA-based treatment can be divided into miRNA mimics (leading to the restoration of miRNA function) and miRNA repressors or inhibitors (leading to downregulation of the target miRNA) [76]. There are several delivery systems for miRNAs to the tumor site, which can be divided into local (i.e., via direct injection to tumor sites or topical delivery to the skin) and systemic delivery systems (i.e., via viral vectors, nanoparticles or exosomes) [76]. To date, therapeutic delivery of miRNAs to cancer patients has faced relevant challenges since several hurdles must be overcome to achieve the desired therapeutic effect (i.e., penetration into the tumor tissue, prevention of degradation in the bloodstream, cellular uptake of the miRNA, prevention of off-target effects, etc.). The following subsections summarize clinical and preclinical experience with the therapeutic delivery of miRNAs.

### 4.1. Clinical Trials Using miRNA Based Approaches

Up to now, only a few phase I clinical trials have tested the therapeutic delivery of miRNAs to cancer patients with modest activity at best. As an example, Beg et al. reported on the therapeutic use of a liposomal miR-34a mimic (MRX34) in 47 patients with advanced solid tumors refractory to all standard treatments. MRX34, a 23-nucleotide long, double-stranded, synthetic version of the tumor-suppressor miR-34a encapsulated in a liposomal nanoparticle with a diameter of ~110 nm, was given as intravenous infusion twice weekly. The authors noted relevant dose-limiting grade III renal-, pulmonary- and gastrointestinal toxicity that resolved with supportive measures. One patient experienced a partial remission lasting for 48 weeks, and six patients achieved a stable disease as their best response [77]. In another study, miR-16 was used via delivery in “minicells” targeted to EGFR (called TargomiRs). TargomiRs were administered intravenously to 27 patients with previously treated advanced mesothelioma at increasing doses. A level of 5 × 10^9^ was deemed the maximum tolerated dose (MTD) after several patients experienced relevant cardiac toxicity (i.e., coronary ischemia, stress cardiomyopathy). All patients experienced fever and chills after TargomiR infusion despite premedication with antihistamines, acetaminophen and steroids. Roughly half of the patients reported non-cardiac chest pain after the infusion, probably related to the accumulation of TargomiRs at the tumor site. One patient experienced a partial remission, and fifteen patients achieved a stabilization of their disease [78]. Both trials were terminated after phase I. Other trials with miRNA-based approaches are recruiting at the moment, including cobomarsen, an inhibitor of miR-155, in patients with mycosis fungoides (NCT02580552 clinicaltrials.gov). Cobomarsen has potent activity against mycosis fungoides [79] and diffuse-large B cell lymphoma (DLBCL) [80] cells in vitro as it leads to decreased cell proliferation and increased apoptosis via inhibition of multiple signaling pathways including JAK/STAT, PI3K/AKT and MAPK. These pathways also regulate immune-checkpoints; however, the effect of cobomarsen on immune-checkpoint molecules has not been reported yet.

Taken together, the clinical trial data are somewhat discouraging so far; however, relevant translational observations have been reported in these trials. As such, Beg et al. noted that the observed toxicity with MRX34 (i.e., pneumonitis, colitis) showed similarity with immune-related adverse events (irAEs) during ICI therapy [77], and Van Zandwijk et al. reported an inverse relationship between the efficacy of miR-16 and PD-L1 expression on mesothelioma cells and speculated whether the response to ICIs could be augmented by the combination with a miR-16 mimic [78].

Such combination approaches have not reached the clinical trial state yet; however, there is growing preclinical evidence for certain potential synergisms between immune-checkpoint blockade and miRNAs, as outlined in the section below.

### 4.2. Preclinical Investigations Using miRNA-Based Approaches

#### 4.2.1. Mesothelioma

The combination of ipilimumab and nivolumab has recently been approved for the treatment of advanced mesothelioma [81]. Kao et al. have previously analyzed miRNA expression profiles of PD-L1 positive and negative mesothelioma tumor samples. They found that the median miRNA expression levels of miR-15b, miR-16, miR-193a-3p, miR-195, and miR-200c were significantly lower in PD-L1–positive samples. Notably, these patients had shorter OS than patients with lower PD-L1 expression. When they performed transfection of mesothelioma cells with miR-15 and miR-16 mimics in vitro, a restoration of PD-L1 expression to baseline was noted, suggesting a potential synergism with ICIs [82].

miR-16 also has been reported to have immune-stimulatory effects on macrophages and T cells. As an example, Jia et al. virally transfected healthy mouse peritoneal macrophages with miR-16. Transfection with miR-16 led to M1 macrophage differentiation and subsequent CD4+ T cell activation via downregulation of PD-L1, suggesting a potential ability of miR-16 to shift macrophages towards a more antitumor phenotype, making combination strategies with ICIs a potentially effective approach [83]. These results also indicate that miR-16 has different functions in tumor cells and cells of the microenvironment underlining the need for specific delivery to the target cell population.

#### 4.2.2. Melanoma

ICIs have been approved in melanoma for several years and have yielded outstanding clinical responses, rendering this disease a good model for evaluating ICI combinations. As such, Xi et al. evaluated the role of miR-21 on macrophages in the tumor microenvironment in a melanoma mouse model [84]. They injected melanoma cell lines into miR-21 deficient and WT mice and found that the knock out mice developed smaller tumors than their WT counterparts. Interestingly, bone marrow transplantation of miR-21 deficient mice into WT mice resulted in similar decreased tumor growth. When they analyzed differences in immune cell infiltration, they noted similar CD4+, CD8+ T cell, and macrophage infiltrates; however, tumors from miR-21 deficient mice displayed more M1 tumor-associated macrophages (TAMs), indicating that these mice displayed a shift towards an antitumor microenvironment. On a molecular level, miR-21 downregulates STAT1 inhibiting IFN-γ induced STAT1 signaling, which leads to M2 polarization. Treatment of the miR-21 deficient mice with an anti-PD-1 mAb led to profound tumor inhibition compared to WT tumors. These data indicate that inhibition of miR-21 in combination with ICIs leads to relevant tumor shrinkage via M1 macrophage polarization in a melanoma mouse model. The necessity of maintained or restored IFN-γ signaling for the success of ICI has been corroborated by Mastroianni et al. In a melanoma xenograft model, the authors found higher IFN-γ levels due to STAT1 activation in miR-146a deficient mice. Mice receiving combined miR-146a antagomir and anti-PD-1 therapy displayed a longer survival compared to mice only receiving anti-PD-1 therapy or the isotype control antibody [85].

Further in vitro evidence of potential combination strategies was reported by Li et al. They performed microarray-based profiling of PD-1 positive and negative CD4+ T cells of melanoma-bearing mice. They found a differential expression of several miRNAs, including miR-28. miR-28 was shown to bind directly to several checkpoint receptors (PD-1, TIM-3, B- and T-lymphocyte attenuator (BTLA)). Transfection of CD4+ T cells with a miR-28 mimic restored T cell function as evidenced by restoration of cytokine production and treatment with a miR-28 inhibitor increased T cell exhaustion [86].

On the contrary, Huffaker et al. explored the role of miR-155 in T cells in a melanoma mouse model. They injected syngeneic melanoma cells into miR-155 knock-out mice and observed increased tumor growth compared with controls. When they assessed various immune cells in the microenvironment of the knock-out mice, a decreased level of IFN-γ inducible genes in macrophages and reduced levels of IFN-γ producing T cells were noted, indicating defective T cell-mediated tumor immunity. Interestingly, these effects were reversible when treating the knock-out mice with different ICIs. ICIs reduced the expression of miR-155 target genes in tumor-infiltrating lymphocytes. Overall, miR-155 was shown to promote the crosstalk between T cells and macrophages in the microenvironment leading to an M1 phenotype typically associated with ICI responses [87]. The authors speculated that the combination of a miR-155 mimic with ICIs might increase treatment responses.

#### 4.2.3. NSCLC

Zhang et al. investigated the interaction of circular RNA fibroblast growth factor receptor 1 (circFGFR1) and miR-381-3p in NSCLC cell lines as well as in mice xenograft models. The authors could show that circFGFR1 upregulates chemokine receptor 4 (CXCR4) expression by sponging miR-381-3p. Mice with high circFGFR1 expressing xenograft lung tumors treated with anti-PD-1 therapy displayed a statistically significantly worse survival [88]. In line with the latter findings, CXCR4 has been shown to play a crucial role in resistance to ICIs in preclinical studies [89,90]. Based on their findings, Zhang et al. suggested therapeutically targeting the circFGFR1/miR-381-3p/CXCR4-related pathway in NSCLC patients undergoing immune-checkpoint blockade.

#### 4.2.4. Glioblastoma

Glioblastoma is one of the hardest to treat tumor entities, and therapeutic responses to checkpoint inhibitors have been low in clinical trials so far [91]. Therefore, augmentations of treatment responses are eagerly awaited. Wei et al. treated glioblastoma-bearing mice with a miR-124 mimic and observed marked tumor shrinkage in vivo. Mechanistically, miR-124 was shown to inhibit STAT3 leading to miR-21 inhibition. T cells in the microenvironment of miR-124 treated mice showed an increase in IFN-γ and tumor necrosis factor (TNF)-α production. Noteworthy, miR-124 had no therapeutic effect in immune-incompetent mice or mice with depleted CD4+ or CD8+ T cells, suggesting an underlying immune-mediated mechanism [92].

Wei et al. also tested the therapeutic effect of miR-138 in a glioma mouse model. After screening several potential miRNA candidates, miR-138 was predicted to bind to PD-1 and CTLA-4 in silico. This was proven in vitro by luciferase assays as miR-138 was shown to bind to the 3-UTR of PD-1 and CTLA-4. Intravenous treatment of glioblastoma bearing mice with a miR-138 mimic led to relevant tumor shrinkage and improved survival time compared to controls. This effect was only seen in immunocompetent mice as miR-138 led to a marked decrease of PD-1, CTLA-4 and FoxP3 in the tumor microenvironment in vivo [93]. The latter findings are in line with the function of miR-138-5p as a repressor of PD-L1 expression in CRC [94]. Taken together, these results suggest miR-124 and miR-138 as a potential combination partner for ICIs.

#### 4.2.5. Head and Neck Squamous Cell Carcinoma (HNSCC)

In head and neck squamous cell carcinoma (HNSCC), ICI treatment is approved in the metastatic setting as second-line treatment [95,96] as well as first-line treatment in combination with chemotherapy and as monotherapy in patients with a PD-L1 CPS ≥ 1 [36]. Yu et al. investigated the role of the let-7 family of miRNAs in HNSCC. The let-7a/b miRNAs were shown to be downregulated in PD-L1 positive tumors, and patients with low let-7a/b expression had an inferior prognosis. Mechanistically, let-7a/b was shown to inhibit PD-L1 glycosylation leading to PD-L1 downregulation. The authors then applied let-7a/b mimics in combination with a CTLA-4 mAb in an HNSCC mouse model. The combination led to the most profound reduction in tumor mass. Analysis of the tumor samples revealed that the combination led to marked CD8+ T cell infiltration and profound IFN-γ production in tumor-infiltrating lymphocytes indicating enhanced immune activity and a potential synergism [97]. These results indicate that the let-7 family of miRNAs has potent activity in the microenvironment by shifting immune cells towards an antitumor phenotype often required for successful ICI treatment.

#### 4.2.6. Breast Cancer

ICIs have been mainly explored in triple-negative breast cancer (TNBC) and have been approved in the metastatic setting for PD-L1 positive tumors based on the immune cell (IC) score [98,99].

Zhang et al. evaluated the role of miR-149-3 in a TNBC mouse model. They analyzed CD8+ spleen T cells from TNBC bearing mice and noted relevant expression of exhaustion markers (i.e., LAG-3, TIM-3, PD-1). When they transfected the CD8+ T cells with a miR-149-3 mimic, they noted a reversal of T cell exhaustion with increased secretion of effector cytokines (i.e., TNF-α, IFN-γ, interleukin (IL)-2) leading to tumor cell killing in vitro, rendering this miRNA a potential candidate for therapeutic interventions [100].

Huang et al. also evaluated the role of let-7b mimics in breast cancer. They used a carrier system with affinity to the mannose receptor responsive to acidic environments, thereby delivering the miRNA mimic specifically to tumor-associated macrophages. In a breast cancer mouse model, delivery of this miRNA mimic led to a reversal of the tumor-suppressive properties of TAMs by acting as a TLR-7 agonist and suppressing IL-10 production, thereby inhibiting tumor growth [101].

#### 4.2.7. Lymphoma

Zheng et al. evaluated the role of miR-155 in DLBCL. They observed that higher serum concentrations of miR-155 were associated with inferior treatment outcomes in patients treated with R-CHOP. To evaluate the mechanism behind this observation, they transfected Epstein–Barr-virus (EBV)-positive lymphoma cells with a miR-155 inhibitor and EBV negative lymphoma cells with a miR-155 mimic and co-cultured these cells with immune cells. Treatment of EBV positive cells with the miR-155 inhibitor led to an increased CD8+ T cell count and inhibited CD8+ T cell apoptosis. The opposite effect was seen in the cells treated with the miR-155 mimic. They could show that miR-155 binds to the 3-UTR region of PD-L1, leading to enhanced gene expression. Interestingly, cells treated with the miR-155 mimic showed increased sensitivity to anti-PD-1 and anti-PD-L1 antibodies. They concluded that miR-155 has oncogenic potential in DLBCL on one hand but may also augment the sensitivity to ICIs [102].

#### 4.2.8. Prevention of Immune-Related Adverse Events

Treatment with ICIs has resulted in unprecedented treatment responses across several tumor types. However, some patients experience considerable immune-related side effects, which can prevent further treatment with ICIs. Marschner et al. investigated the role of miR-146a in the context of ICI therapy. Mice with knocked-out miR-146a developed severe irAEs after treatment with ICIs due to increased T cell activation and effector functions. It is noteworthy that patients with a single-nucleotide-polymorphism (SNP) in the miR-146a gene were more likely to develop severe irAEs and showed decreased PFS after treatment with ICIs. Interestingly, therapeutic delivery of a miR-146a mimic to the mice led to a reduction in the severity of irAEs, indicating a therapeutic potential to “fine-tune” ICI therapy [103].

Taken together, these results indicate that miRNAs have the potential to augment responses to ICI treatment, mostly via effects on the tumor microenvironment. However, different miRNAs regulate several checkpoint molecules and may even have opposite roles depending on the cell in which they are expressed. Therefore, several hurdles must be overcome before miRNA-based treatment can be applied in cancer patients in a safe and precise way.

## 5. Conclusions

Dysregulated miRNA expression plays a crucial role in cancer and is caused/influenced by several mechanisms such as genetic aberrations at chromosomal loci encoding miRNA genes, cancer-specific expression of transcription factors, epigenetics, and RNA-editing as shown in various malignancies.

Therapeutically targeting immune-checkpoints by ICIs has revolutionized cancer therapy in recent years. However, only a minority of biomarker-unselected patients respond to ICIs and only a few predictive—mainly tissue-based—biomarkers such as PD-L1 expression and TMB have been established so far in clinical practice. In search of predictive biomarkers, miRNAs collected from plasma or EVs provide the advantage of avoiding invasive procedures as well as circumventing the issue of intra- and intertumoral heterogeneity. Although multiple reports have shown the potential of miRNAs to predict clinical outcomes in cancer patients undergoing ICI therapy, several issues such as heterogeneity of applied methodologies, the impact of prior systemic therapies on miRNA expression levels, as well as the small sample size of previous reports represent major drawbacks. MiRNAs impacting on the efficacy of ICIs have been shown to regulate, e.g., macrophage polarization (miR-21, let-7a/b, miR-155), cytokine secretion (e.g., IFN-γ or TNF-α: miR-124, miR-146a, miR-28) as well as T cell exhaustion (miR-28, miR-138, miR-149-3). MiRNA expression levels associated with a beneficial or unfavorable clinical outcome during immune-checkpoint blockade are summarized in detail in Table 1 and Table 2. Besides response prediction, miRNAs also harbor the potential to predict ICI toxicity, as it has been shown for miR-146a. Prospective validation in clinical trials and a comparison with the performance of established biomarkers such as, e.g., tissue-based PD-L1 expression or TMB will be prerequisites before the implementation of plasma-based miRNAs or miRNA panels as predictors of response to ICIs. Unfortunately, the few clinical phase I trials investigating miRNA-based therapy (e.g., miR-34a, miR-16) in cancer patients only showed limited efficacy, but clinically relevant toxicities precluding initiation of clinical phase II trials so far. However, findings from a plethora of preclinical studies are hypothesis-generating and provide the rationale for combining miRNA-based therapy with ICIs in order to enhance ICI efficacy or to overcome resistance to the immune-checkpoint blockade.

## Figures and Tables

**Table 1 pharmaceuticals-14-00089-t001:** Studies investigating microRNAs (miRNAs) as predictors of clinical outcome during the immune-checkpoint blockade.

Author	Year	Tumor Type	Patients	ICI	ICI Therapy Line	miRNAs	Validation	Outcome
Halvorsen et al. [13]	2018	NSCLC (whole plasma)	20	anti-PD-1	≥2nd line	miR-215-5p  , miR-411-3p (NA), miR-493-5p  , miR-494-3p  , miR-495-3p  , miR-548j-5p  , miR-93-3p 	yes	OS
Peng et al. [48]	2019	NSCLC (plasma EV)	16	anti-PD-1, anti-PD-L1	≥1st line	miR-320d  , miR-320c  , miR-320b  , miR-125-5p 	no	response
Boeri et al. [14]	2019	NSCLC (whole plasma)	111	anti-PD-1, anti-PD-L1, anti-PD-L1+ anti-CTLA-4	≥1st line	miR-101-3p, miR-106a-5p, miR-126-5p, miR-133a, miR-140-3p, miR-140-5p, miR-142-3p, miR-145-5p, miR-148a-3p, miR-15b-5p, miR-16-5p, miR-17-5p, miR-197-3p, miR-19b-3p, miR-21-5p, miR-221-3p, miR-28-3p, miR-30b-5p, miR-30c-5p, miR-320a, miR-451a, miR-486-5p, miR-660-5p, miR-92a-3p(miRNA ratios were used to obtain miRNA signatures and in turn to calculate the MSC risk: low, intermediate and high [75])	no	response, PFS, OS
Genova et al. [53]	2020	NSCLC(plasma EV)	174	anti-PD-1	≥2nd line	miR-208a-5p  ,miR-574-5p 	yes	OS
Shukuya et al. [47]	2020	NSCLC(whole plasma and plasma EV)	29	anti-PD-1, anti-PD-L1	NA	miR-548am-5p  , miR-200a-3p  , miR-4707-3p  , miR-335-3p  , miR-429-3p  , miR-200b-3p  , miR-191-3p  , miR-1277-3p  , miR-200c-3p  , miR-28-5p  , miR-3120-3p  , miR-152-3p  , miR-335-5p  , miR-199a-1-3p  , miR-22-5p  , miR-30e-3p  , miR-33a-5p  , miR-556-5p  , miR-21-3p  , miR-30d-3p  , miR-130b-5p  , miR-24-1-3p  , miR-3138-3p  , miR-548ax-5p  , miR-6791-3p  , miR-1287-5p  , miR-3074-5p  , miR-103a-1-3p  , miR-21-5p  , miR-130b-3p  , miR-186-5p  , miR-660-3p  , miR-1246-5p  , miR-1296-5p  , miR-4707-3p  , miR-1229-3p  , miR-874-3p  , miR-378c-5p  , miR-1468-5p 	yes	response
Sudo et al. [62]	2020	ESSC(whole plasma)	19	anti-PD-1	≥2nd line	miR-1233-5p  , miR-6885-5p  , miR-4698  , miR-128-2-5p 	no	response
Miyamoto et al. [64]	2018	GC(whole plasma and plasma EV)	20	anti-PD-1	≥3rd line	two miRNAs termed “miR-A”  and “miR-B”  (not further specified by authors)	no	response, PFS
Huber et al. [15]	2018	Melanoma(whole plasma and plasma EV)	49	anti-PD-1, anti-CTLA-4	≥1st line	let-7e  , miR-125a  , miR-99b  , miR-146b  , miR-125b 	no	OS, PFS
Galore-Haskel et al. [74]	2015	Melanoma(tumor tissue)	13	anti-CTLA-4	≥1st line	miR-222 	yes	response

CTLA-4: cytotoxic T-lymphocyte protein 4, ESSC: esophageal squamous-cell carcinoma, EV: extracellular vesicles, GC: gastric cancer, ICI: immune-checkpoint inhibitor, MSC: microRNA signature classifier, NA: not available, NSCLC: non-small cell lung cancer, OS: overall survival, PFS: progression-free survival, PD-1: programmed cell death protein 1, PD-L1: programmed death ligand 1. Arrows indicate miRNA expression levels, and color code depicts associated clinical outcomes (green: better clinical outcome, red: worse clinical outcome).

**Table 2 pharmaceuticals-14-00089-t002:** In vitro and in vivo studies investigating miRNAs as therapeutic substances.

miRNA	Tumor Type	Experimental Setting	Outcome	Author
miR-16	Healthy tissue	Mouse model	M1 macrophage differentiation, T cell activation and downregulation of PD-L1 following viral miR-16 transfection	Jia et al. [83]
	Mesothelioma	Humans (phase I)	3% PR, 55% SD with a miR-16 based mimic	Van Zandwijk. [78]
miR-21	Melanoma	Mouse model	Inhibition of M2 macrophage differentiation and increased tumor cell killing in combination with ICI in miR-21 deficient mice	Xi et al. [84]
miR-28	Melanoma	Mouse model	Restoration of T cell function and increased cytokine production following transfection with a miR-28 mimic	Li et al. [86]
miR-34a	Advanced tumors	Humans (Phase I)	2% PR, 12% SD with a liposomal miR-34a mimic	Beg et al. [77]
miR-124	Glioblastoma	Mouse model	Increased IFN-γ production by T cells following treatment with a miR-124 mimic	Wei et al. [92]
miR-138	Glioblastoma	Mouse model	Decreased expression of T cell exhaustion markers following treatment with a miR-138 mimic	Wei et al. [93]
	CRC	Mouse model	PD-L1 downregulation by miR-138-5p	Zhao et al. [94]
miR-146a	Melanoma	Mouse model	Increased IFN-γ expression levels in miR-146a deficient mice Increased ICI sensitivity with combined miR-146a antagomiR treatment	Mastroianni et al. [85]
miR-149-3	TNBC	Mouse model	Reversal of T cell exhaustion and increased secretion of effector cytokines after transfection with a miR-149-3 mimic	Zhang et al. [100]
miR-155	Melanoma	Mouse model	miR-155 triggers M1 macrophage differentiation	Huffaker et al. [87]
	DLBCL	In vitro	Increased sensitivity to ICI after transfection with a miR-155 mimic	Zheng et al. [102]
miR-381-3p	NSCLC	Mouse model	Decreased CXCR4 expression and increased ICI sensitivity by miR-381-3p	Zhang et al. [88]
let-7a/b	HNSCC	Mouse model	CD8+ T cell infiltration and cytokine production by combining let-7a/b mimics with ICI	Yu et al. [97]
	TNBC	Mouse model	Reversal of tumor-suppressive properties of TAMs following delivery of let-7b mimics	Huang et al. [101]

CRC: colorectal cancer, DLBCL: diffuse large B cell lymphoma, HNSCC: head and neck squamous cell carcinoma, ICI: immune-checkpoint inhibitor, IFN-γ: interferon-γ, miR: microRNA, NSCLC: non-small cell lung cancer, PR: partial remission, SD: stable disease, TAMs: tumor-associated macrophages, TNBC: triple-negative breast cancer.

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
