# Peer review of "miRNA-Based Therapeutics in the Era of Immune-Checkpoint Inhibitors"

_pharmaceuticals, 2021, doi:10.3390/ph14020089_

Round 1
Reviewer 1 Report
The manuscript entitled “miRNA-based Therapeutics in the Era of Immune-Checkpoint Inhibitors” by Florian Huemer et al. needs to be improved upon consideration of the following points
1) Since authors demonstrated the role of miRNAs in response prediction in section 3, it is better to generally describe the role of miRNAs as biomarkers in introduction or section 2.
2) Canonical functions of miRNAs are well described in introduction section. However, authors also need to mention about their non-canonical functions.
3) For the description of miRNA functions, authors need to be more critical. For example, although miR-320 is described as tumor suppressive miRNA in lung cancer, another evidence shows that miR-320 promote M2-macrophages (reference: Circulating mir-320a promotes immunosuppressive macrophages M2 phenotype associated with lung cancer risk. Int J Cancer. 2019 Jun 1;144(11):2746-2761).
4) For Section 4, please provide Table 2 to efficiently summarize contents (including in vitro and in vivo experiments).
5) It is required to denote if miRNAs are low or high in good responders using arrows (Table 1).
6) Lines 152, 153, 187: miR-215 (miR-215-5p?) / miR548j (-5p? and hyphen is missing) / interferon-(IFN) (delete hyphen). miR320a (hyphen is missing). Carefully check the whole manuscript and correct typos etc if it is necessary.
7) In Table 1, section 3, and other parts: use miR- or hsa-miR throughout the manuscript.
Reviewer 2 Report
This paper by F. Huemer et.al reviewed the role of miRNAs in cancer patients treated with immune-checkpoint inhibitors. Howerer, the authors just list the results of each paper one by one. The authors need to summarize these papers and let the readers know the status of miRNA studies in ICI.
- The authors reviewed the role of miRNAs in different types of cancers. The authors need to summarize some key miRNAs that are good or not good for ICI treatment.
- There are some studies not cited in this paper. I just list some of them and the authors need to re-search the related papers. 1) Zhang PF, Pei X, Li KS, Jin LN, Wang F, Wu J, et al. Circular RNA circFGFR1 promotes progression and anti-PD-1 resistance by sponging miR-381-3p in non-small cell lung cancer cells. Mol Cancer. 2019;18:179. 2) Zhao L, Yu H, Yi S, Peng X, Su P, Xiao Z, et al. The tumor suppressor miR-138-5p targets PD-L1 in colorectal cancer. Oncotarget. 2016;7:45370–84. 3) Mastroianni J, Stickel N, Andrlova H, Hanke K, Melchinger W, Duquesne S, et al. miR-146a Controls Immune Response in the Melanoma Microenvironment. Cancer Res. 2019;79:183–95.
Round 2
Reviewer 1 Report
The authors have significantly improved the manuscript in an impressively short time. All required corrections have been included and the confusion about some structure descriptions has been resolved with appropriate terminology. I do not have any further concerns.
Reviewer 2 Report
Most of my concerns are addressed. Agree to publish.